# Effects of iNOS in Hepatic Warm Ischaemia and Reperfusion Models in Mice and Rats: A Systematic Review and Meta-Analysis

**DOI:** 10.3390/ijms231911916

**Published:** 2022-10-07

**Authors:** Richi Nakatake, Mareike Schulz, Christina Kalvelage, Carina Benstoem, René H. Tolba

**Affiliations:** 1Institute for Laboratory Animal Science and Experimental Surgery, RWTH Aachen University, 52074 Aachen, Germany; 2Department of Surgery, Kansai Medical University, 2-5-1 Shinmachi, Hirakata, Osaka 573-1010, Japan; 3Department of Intensive Care Medicine, Medical Faculty, RWTH Aachen University, 52074 Aachen, Germany

**Keywords:** warm ischaemia, reperfusion, iNOS, liver, rodent

## Abstract

Warm ischaemia is usually induced by the Pringle manoeuver (PM) during hepatectomy. Currently, there is no widely accepted standard protocol to minimise ischaemia-related injury, so reducing ischaemia-reperfusion damage is an active area of research. This systematic review and meta-analysis focused on inducible nitric oxide synthase (iNOS) as an early inflammatory response to hepatic ischaemia reperfusion injury (HIRI) in mouse- and rat-liver models. A systematic search of studies was performed within three databases. Studies meeting the inclusion criteria were subjected to qualitative and quantitative synthesis of results. We performed a meta-analysis of studies grouped by different HIRI models and ischaemia times. Additionally, we investigated a possible correlation of endothelial nitric oxide synthase (eNOS) and nitric oxide (NO) regulation with iNOS expression. Of 124 included studies, 49 were eligible for the meta-analysis, revealing that iNOS was upregulated in almost all HIRIs. We were able to show an increase of iNOS regardless of ischemia or reperfusion time. Additionally, we found no direct associations of eNOS or NO with iNOS. A sex gap of primarily male experimental animals used was observed, leading to a higher risk of outcomes not being translatable to humans of all sexes.

## 1. Introduction

Hepatectomy is a well-established therapeutic option for benign and malignant liver disease. [1,2,3]. A total of 21,443 patients underwent hepatectomy between 2012 and 2016 in the United States (American College of Surgeons National Surgical Quality Improvement Program’s Participant Use Files; male 10,476, female 10,967), 28,708 (male 15,489 and female 13,219) between 2007 and 2010 in France (the Medicalization des Systemes D’Information databases), 110,332 (male 55,781 and female 54,551) between 2010 and 2015 in Germany (Diagnosis-Related Groups; ICD-10 and German operations and procedure key codes) and 20,575 in 2011 in Japan (the National Clinical Database of Japan) [4,5,6,7].

Hepatectomy is a complicated abdominal surgical technique associated with a prolonged surgical time and an increased risk of perioperative complications, such as bleeding. To minimize bleeding, the Pringle manoeuver (PM), a clamping technique for the portal triad, is used intermittently or continuously and enables nonselective inflow occlusion by controlling blood loss during hepatectomy [8,9,10]. However, these techniques also simultaneously induce warm ischaemia. A recent survey in Europe showed that 71% of surgeons apply PM on indication, whereas 19% use PM routinely. Only 10% of surgeons never use PM, although it is the most frequently used technique in hepatectomy [11]. Despite the huge importance of this technique, only three randomised human trials have been conducted to compare hepatectomy with and without intermittent PM [12,13,14]. These trials reported conflicting results, which has led to even more discussion on the value of PM.

Hepatic ischaemia-reperfusion injury (HIRI) can occur during interventions, causing direct hepatic ischaemia in liver surgery or transplantation. HIRI is a serious clinical condition leading to cellular injury and organ dysfunction, mainly through production of reactive oxygen species and inflammatory cytokines [15]. The implications of warm HIRI are apoptosis of hepatocytes and sinusoidal endothelial cells, necrosis or a combination of both [16]. During hepatic ischaemia-reperfusion (HIR), there are interactions among liver cells, Kupffer cells, neutrophils, hepatic sinusoidal endothelial cells, and fat-storing cells. Platelets and alexin are also involved [17]. These activated cells release a large quantity of proinflammatory cytokines and lipid inflammatory factors, which can lead to further inflammatory reactions and cell apoptosis. Preventing HIRI is, therefore, important for regeneration after liver operation.

The function of nitric oxide (NO) in HIRI is inconclusive and complex. Whether or not NO has cytoprotective or cytotoxic effects is controversial because it has various roles throughout the body. It is recognised that NO maintains a continuous balance between vasodilatation and vasoconstriction due to continuous production [18]. In this context, NO and its derivatives are known to have important roles in the pathophysiology of the liver [19]. NO has been proven to reduce HIRI through various mechanisms [20,21]; and it is an unstable nitrogen-centred radical produced in a redox reaction between L-arginine and oxygen molecules by NO synthase (NOS) catalysation. There are three types of NOS: endothelial nitric oxide synthase (eNOS), neuronal nitric oxide synthase (nNOS) and inducible nitric oxide synthase (iNOS). In many cell types, including liver endothelial cells and hepatocytes, eNOS is constitutively expressed [21,22]. In contrast, iNOS is expressed under pathological circumstances, such as endotoxemia, haemorrhagic shock, hepatitis and liver regeneration, but is upregulated in various nucleated cell types, including hepatocytes, neutrophils, T-lymphocytes, endothelial, biliary and Kupffer cells, under inflammatory conditions associated with HIRI [23,24,25,26,27]. In general, it is assumed that NO derived from eNOS in liver sinusoidal endothelial cells is cell-protective, whereas iNOS-derived NO contributes to pathological processes by acting as a proinflammatory mediator [4]. NO is produced mainly by eNOS catalysis and upregulation of iNOS expressions during acute warm HIR [28]. An excess level of NO produced by iNOS has been implicated as a factor in liver injury [29]. iNOS is induced to produce large amounts of NO by lipopolysaccharides and proinflammatory cytokines, such as interleukin-1 (IL-1) and tumour necrosis factor (TNF), which have a role in many inflammatory and immune reactions [30]. During early warm HIRI, NO produced by eNOS activation is protective, but NO induced by iNOS may have either a protective or deleterious effect [31]. Therefore, strategies that target deleterious NO effects comprise an active area of discussion as potential new therapies, which have been experimentally investigated in many disease models. Animal-based research may give insights into the mechanisms of iNOS in the HIRI model, but the knowledge gained from most single studies is limited. It remains unclear how NO induced by iNOS is regulated during warm HIRI. Furthermore, the relationships between iNOS and eNOS in HIRI could not be clarified in detail. Therefore, a systematic review addressing this specific problem may provide more detailed insight into this topic and provide information about various experimental results.

## 2. Results

### 2.1. Results of the Search

The literature search resulted in 793 records. After removing 365 duplicates, 428 records remained, and their titles and abstracts were screened. We excluded 279 records that did not meet the pre-specified inclusion criteria and screened the full texts of the remaining 149 references. Twenty-five records were excluded after full-text assessment, and the reasons for exclusion are shown in Figure 1. In total, we included 124 records in our synthesis, from which 49 records were also used in the meta-analysis. The search process is described fully in the flow diagram in Figure 1.

### 2.2. General Characteristics

The characteristics of the 124 included studies are presented in Table 1, Table 2, Table 3, Table 4 and Table 5. The studies were classified according to the models of the study.

### 2.3. Details for Rat Models

Eighty-nine studies containing rat models were identified. Fifty-six (63%) studies with 70% HIRI were reported (Table 1). The ischaemic time taken varied from 5 to 120 min, and most of the studies used 45 (6 studies), 60 (25 studies) or 90 min (9 studies). In these studies, the reperfusion time varied between 0.5 and 168 h. For other studies with different or unknown ischaemic times, the number of results by subgroup was too small (*n* < 3) for a meta-analysis.

**Table 1 ijms-23-11916-t001:** Overview of rat models with 70% HIRI.

Reference	Year	Strain	Reperfusion Time (h)	Gender (%)	BW (g)	iNOS Detection Method	NO Detection	eNOS Detection	Survival Rate
**30 min ischemia time**									
Abd-Elbaset [32]	2017	*Wistar*	0.5	♂ 100%	300–350	IHC	-	+	-
Fouad [33]	2011	*SD*	72	♂ 100%	190–210	IHC	+	-	-
Liu [34]	2000	*Fischer*	4	♂ 100%	275–300	RNA	+	-	-
Liu [35]	1998	*Fischer*	4	♂ 100%	245–290	SP	-	+	-
Rhee [36]	2002	*SD*	1/6	♂ 100%	100–150	indirect	+	-	+
Wang [37]	2017	*SD*	1	♂ 100%	200–250	Protein	-	-	-
Wang [38]	2003	*Wistar*	6/1/3/5/7/336	♂ 100%	240–260	RNA/WB	-	+	-
**45 min ischemia time**									
El-Emam [39]	2020	*SD*	24	♂ 100%	250–280	ELISA	-	-	-
Hur [40]	1999	*SD*	0.5/1/3/5/12/24	♂ 100%	350–400	RNA	-	-	-
Koti [21]	2005	*SD*	2	♂ 100%	250–300	WB	+	+	-
Mostafa-Hedeab [41]	2019	*Wistar*	N/A	♂ 100%	140–250	RNA	-	+	-
Serracino-Iglott [42]	2003	*Wistar*	1	♂ 100%	250–300	WB/IHC	-	+	-
Yang [43]	2011	*SD*	2	♂ 100%	250–300	IHC	+	+	-
**60 min ischemia time**									
Curek [44]	2010	*Wistar*	1	♂ 100%	350–450	IHC	-	-	-
Eum [45]	2004	*SD*	5	♂ 100%	270–300	RNA	-	+	-
Eum [46]	2004	*SD*	5/24	♂ 100%	260–300	RNA	-	-	-
Fernández [47]	2009	*SD*	0.33	♂ 100%	180–200	RNA	-	-	-
Ferrigno [48]	2020	*Wistar*	1	♂ 100%	N/A	WB	-	-	-
Ferrigno [49]	2020	*Wistar*	1/2	♂ 100%	N/A	WB	-	+	-
Hataji [50]	2010	*SD*	24	♂ 100%	N/A	IHC	+	+	-
Hsu [51]	2002	*SD*	2	♂/♀ 50%	200–275	IHC/activity	-	-	-
Kang [52]	2011	*SD*	1/5	♂ 100%	270–300	RNA/WB	-	-	-
Kim [53]	2012	*SD*	5	♂ 100%	150–170	WB/RNA	-	-	-
Kim [54]	2010	*SD*	5	♂ 100%	270–300	RNA/WB	-	-	-
Kim [55]	2004	*SD*	5	♂ 100%	260–320	RNA	-	+	-
Kurabayashi [56]	2005	*SD*	5	♂ 100%	230–290	IHC	-	-	-
Lee [57]	2008	*SD*	5	♂ 100%	270–300	RNA	-	+	-
Man [58]	2005	*SD*	0.33/1/1.5/6/24	♂ 100%	220–280	RNA	-	-	-
Park [59]	2007	*SD*	5	♂ 100%	270–300	RNA	-	+	-
Ramalho [60]	2014	*Wistar*	6	♂ 100%	150–200	WB	+	+	-
Ren [61]	2019	*Wistar*	1/3/6/12	♂ 100%	220–280	RNA	-	-	-
Sonin [62]	1999	*SD*	6	♂ 100%	250–330	RNA	-	-	-
Tao [63]	2014	*Wistar*	6	♂ 100%	180–220	ELISA	-	-	+
Trocha [64]	2014	*Wistar*	4	♂ 100%	N/A	ELISA	-	-	-
Unal [65]	2017	*Wistar*	1	♂ 100%	350–450	IHC	+	-	-
Yang [66]	2007	*SD*	1/3/5	♂ 100%	230–250	mRNA	+	+	-
Yun [67]	2012	*SD*	5	♂ 100%	270–300	RNA/WB	-	-	-
Yun [68]	2010	*SD*	1/2/4/6/8/12/24	♂ 100%	270–300	RNA/WB	-	-	-
**90 min ischemia time**									
Bektas [69]	2016	*Wistar*	2	♂ 100%	250–300	IHC	-	+	-
Grezzana-Filho [70]	2020	*Wistar*	24	♂ 100%	250–310	WB	-	+	-
Kim [71]	2004	*SD*	6	♂ 100%	260–300	RNA	-	+	-
Kuncewitch [72]	2013	*SD*	24	♂ 100%	250–275	WB	-	-	+
Lin [73]	2004	*Wistar*	1.5	♂ 100%	300–350	IHC	+	+	-
Longo [74]	2016	*Wistar*	2	♂ 100%	200–250	WB	-	+	-
Takamatsu [75]	2006	*SD*	3/6/12/24	♂ 100%	230–300	RNA	+	-	-
Yao [76]	2009	*SD*	1/3/6/24/168	♂ 100%	220–240	Protein	+	-	+
Yun [77]	2012	*SD*	3/24	♂ 100%	270–300	WB	-	-	-
**Studies not included in meta-analysis**									
**20 min ischemia time**									
Wang [78]	1998	*Fischar*	0.5	♂ 100%	240–320	N/A	+	-	-
**35 min ischemia time**									
Kireev [79]	2012	*Wistar*	36	♂ 100%	N/A	RNA	-	+	-
Kireev [80]	2013	*fa/fa Zucker*	36	♂ 100%	496	RNA	-	+	-
**40 min ischemia time**									
Duan [81]	2017	*SD*	2	♂ 100%	190–210	WB	+	+	-
Trocha [82]	2010	*Wistar*	1	♂ 100%	240–303	ELISA	-	+	-
**100 min ischemia time**									
Hara [83]	2005	*N/A*	12	N/A	N/A	WB	-	-	-
**120 min ischemia time**									
Ishizaki [84]	2008	*SD*	1/3/6/9/12/24	♂ 100%	240–270	RNA/WB	+	-	-
**Unknown ischemia time**									
Hsieh [85]	2015	*SD*	3.5/24	♂ 100%	250–300	RNA	-	-	-

ELISA, Enzyme-linked immunosorbent assay; IHC, immune histochemistry; N/A, not available; RNA, ribonuclein acid; SP, spectrophotometric analysis; SD, *Spraque– Dawley*; WB, Western blot; + reported; - not reported.

Figure 2 shows the distribution of warm ischaemia times compared with their reperfusion times for each study of 70% HIRI in rat models.

As shown in Table 2, 22 (24.7%) of the included studies reported a HIRI of 100%. The ischaemic time taken varied from 5 to 120 min, and most of the studies involved 30 (6 studies), 45 (7 studies) or 60 (4 studies) min. In these studies, the reperfusion time varied between 0 and 144 h. For other studies with different ischaemic times, the number of results by subgroup was too small (*n* < 3) for a meta-analysis.

**Table 2 ijms-23-11916-t002:** Overview of rat models with 100% HIRI.

Reference	Year	Strain	Reperfusion Time (h)	Gender	BW (g)	iNOS Detection Method	NO Detection	eNOS Detection	Survival Rate
**30 min ischemia time**									
Acquaviva [86]	2009	*Wistar*	3	♂ 100%	200–220	WB	-	+	-
Chen [87]	2014	*SD*	2	♂ 100%	250–300	-	+	-	-
Lanteri [88]	2007	*Wistar*	0.5/3	♂ 100%	200–220	WB	-	+	-
Morisuee [89]	2003	*Wistar*	3/4/6	♂ 100%	350–450	IHC	+	-	+
Nii [90]	2014	*Wistar*	2	♂ 100%	250–300	RNA	-	-	-
Uchiami [91]	2002	*Wistar*	0.5/1/2	♂ 100%	280–320	RNA	-	-	-
**45 min ischemia time**									
Atef [92]	2017	*Wistar*	1	♂ 100%	200–250	RNA	+	+	-
El-Shintany [93]	2015	*albino*	2	♂ 100%	180–200	IHC	+	-	-
Ibrahim [94]	2020	*albino*	24	♂ 100%	200–300	ELISA	-	+	-
Ibrahim [95]	2014	*albino*	1	♂ 100%	200–230	IHC	+	-	-
Sankary [96]	1999	*SD*	0.25/0.5/1/2/3/144	♂ 100%	200–250	-	-	-	+
Sehitoglu [97]	2019	*Wistar*	24	♂ 100%	250–300	IHC	-	-	-
Yaylak [98]	2008	*Wistar*	0.75	♂ 100%	150–220	IHC	-	-	-
**60 min ischemia time**									
Miyake [28]	2013	*Wistar*	5/3	♂ 100%	250–300	WB/RNA	+	-	+
Rodríguez-Reynoso [99]	2001	*SD*	1/2	♂ 100%	250–300	RNA	+	-	+
Sankary [96]	1999	*SD*	0.25/0.5/1/2/3/144	♂ 100%	200–250	-	-	-	+
Yang [100]	2007	*SD*	5	♂ 100%	230–250	mRNA	+	+	-
**Studies not included in meta-analysis**									
**5/15 min ischemia time**									
Miyake [28]	2013	*Wistar*	5,3	♂ 100%	250–300	WB/RNA	+	+	-
**20 min ischemia time**									
Harada [101]	2003	*SD*	2/6/24	♂ 100%	225–250	RNA	-	-	-
Suetsugu [102]	2005	*SD*	72	♂ 100%	240–255	WB	-	-	-
**40 min ischemia time**									
Abdel-Gaber [103]	2015	*Albino*	1	♂ 100%	200–230	IHC	+	+	-
**90 min ischemia time**									
Koeppel [26]	2007	*SD*	3/6/12/24	♂ 100%	280–350	RNA	-	+	-
Rodriguez-Reynos [104]	2018	*SD*	1/5/168	♂ 100%	250–300	RNA	+	-	+
Xue [105]	2010	*SD*	24/48/72/168	♂ 100%	200–250	RNA	+	+	+

ELISA, Enzyme-linked immunosorbent assay; IHC, immune histochemistry; N/A, not available; RNA, ribonuclein acid; SD, *Spraque–Dawley*; WB, Western blot; + reported; - not reported.

Figure 3 shows the distribution of warm ischaemia times compared with their reperfusion times for each study of 100% HIRI in rat models.

Rat HIRI models with additional procedures, such as ischaemic preconditioning, cirrhotic liver, atrial hepatectomy or splenic–caval shunt, were not included in the meta-analysis because of significant methodological heterogeneity with regard to the experimental model. These references are presented in Table 3, divided by the presence or absence of hepatectomy and ordered by the size of the ischaemia area. In total, seven studies performed partial hepatectomies.

**Table 3 ijms-23-11916-t003:** Overview of other rat models (not included in the meta-analysis).

Model	Reference	Year	Strain	Reperfusion Time (h)	Gender	BW (g)	iNOS Detection Method	NO Detection	eNOS Detection	Survival Rate
30% HIRI 60 min ischemia time	Wang [106]	2012	*SD*	6	♂ 100%	250–300	WB	-	-	-
40% HIRI 45 min ischemia time	Björnsson [107]	2015	*SD*	4	♂ 100%	313–444	RNA/IHC	+	-	-
40% HIRI 60 min ischemia time	Björnsson [108]	2014	*SD*	1/4	♂ 100%	258–444	RNA	+	-	-
32% HIRI + 68% PH 30 min ischemia time	Liang [109]	2009	*SD*	3/8/48/168	♀ 100%	200–220	IHC	-	-	+
50%HIRI + 50%PH 45 min ischemia time	Iwasaki [110]	2019	*Wistar*	1/3/24/168	♂ 100%	200–300	RNA/WB	+	+	-
70% HIRI + 30%PH 45 min ischemia time	Shen [111]	2007	*SD*	3/12/24/168	♂ 100%	220–300	SP	+	+	+
70% HIRI + 30%PH 45 min ischemia time	Shen [112]	2007	*SD*	3/12/24	♂ 100%	250–300	SP	+	+	+
70% HIRI + 30%PH 60 min ischemia time	El-Gohary [113]	2017	*Wistar*	1/5	♂ 100%	200–250	WB	-	+	-
70% HIRI + 30%PH 60 min ischemia time	Zhang [114]	2015	*SD*	6/168	♂ 100%	250–300	WB	+	-	-
70% HIRI + 70%PH 30 min ischemia time	Duval [115]	2010	*SD*	0/0.5/1/3/6/9/12/15/18/21/24/30/48/72	♂ 100%	200–225	RNA	-	-	-
100% HIRI + 70%PH 15 min ischemia time	Kawai [116]	2010	*Wistar*	1	♂ 100%	250–300	RNA/IHC	+	+	-
120, 150 min ischemia time + SCS	Sankary [96]	1999	*SD*	≤144	♂ 100%	200–250	IHC	-	-	-

IHC, immune histochemistry; PH, Partial hepatectomy; RNA, ribonuclein acid; SCS, splenic-caval shunt; SD, *Spraque–Dawley*; SP, Spectrophotometric analysis; WB, Western blot; + reported; - not reported.

### 2.4. Details for Mouse Models

Thirty-nine studies in mouse models were included. As shown in Table 4, 37 (95%) studies with 70% HIRI were reported. The ischaemic time varied from 30 to 90 min. Ischaemic times of 45 (6 studies), 60 (19 studies) and 90 (7 studies) min were included in the meta-analysis.

**Table 4 ijms-23-11916-t004:** Overview of mouse models with 70% HIRI.

Reference	Year	Strain	Reper-fusion Time (h)	Gender	Age (Weeks)	iNOS Detection Method	NO Detec.	eNOS Detec.	Survival Rate
**45 min ischemia time**									
Bae [117]	2014	*C57Bl/6*	1/6/24	♂ 100%	8–10	RNA	-	-	-
Datta [118]	2014	*C57Bl/6 (eNOS −/− and WT)*	2	♂ 100%	8–12	WB	-	+	-
Hines [119]	2001	*C57Bl/6 (iNOS −/− and WT)*	1/3/6	♂ 100%	N/A	RNA	-	-	-
Hines [120]	2002	*C57Bl/6 (iNOS −/−, eNOS −/− and WT)*	1/3	♂ 100%	N/A	RNA	-	+	-
Kawachi [121]	2000	*C57Bl/6 (NOS −/−, eNOS −/− and WT)*	5	♂ 100%	N/A	RNA	-	+	-
Lee [122]	2001	*C57Bl/6 (iNOS −/−, eNOS −/− and WT)*	6	♂/♀ 50%	N/A	RNA	-	-	-
**60 min ischemia time**									
Chen [123]	2017	*C57Bl/6 (NLRC5 −/− and WT)*	N/A	♂ 100%	8–10	ELISA	-	-	-
Gao [124]	2016	*Cav* *-1tm1Mls/J (Cav* *-1 −/−and WT)*	1/6/12	♂ 100%	8–12	WB/ICH	-	-	-
Guo [125]	2011	*BALB/c*	2/4/24	♂ 100%	N/A	SP	+	+	-
Guo [126]	2011	*BALB/c*	2/4/12	♂ 100%	N/A	WB/SP	-	+	-
Jeyabalan [127]	2008	*C57Bl/6*	3/6	♂ 100%	8–12	RNA	+	+	-
Kim [128]	2015	*C57Bl/6*	6	♂ 100%	6–8	RNA	-	-	-
Klune [129]	2012	*C57BL/6 (IRF2 −/−, IRF2 +/− and WT)*	6	♂ 100%	8–12	RNA	-	-	-
Lee [122]	2001	*C57Bl/6 (iNOS −/−, eNOS −/− and WT)*	6	♂/♀ 50%	N/A	RNA	-	-	-
Luedde [130]	2005	*Ikk2 −/−, Nemo −/−*	6/24	♂ 100%	8–10	IHC	-	-	-
Moon [131]	2008	*C57Bl/6*	2	♂ 100%	N/A	WB	-	-	-
Mukhopadhyay [132]	2011	*C57Bl/6*	2/6/24	♂ 100%	N/A	RNA	-	-	-
Qiao [133]	2020	*C57Bl/6 (iNOS −/− and WT)*	6/24	♂ 100%	8–10	WB	-	-	-
Qiao [134]	2019	*C57Bl/6 (iNOS −/− and WT)*	6/24	♂ 100%	~8	RNA/WB	-	-	-
Sanches [135]	2014	*Swiss*	6	♂ 100%	N/A	WB	+	+	-
Shaker [136]	2016	*C57Bl/6*	3/12	♂ 100%	8–10	RNA	-	-	-
Shi [137]	2012	*C57Bl/6*	2/6/168	♂ 100%	8–9	WB	-	-	+
Tsung [138]	2006	*C57Bl/6 (IRF−1 −/− and WT)*	1/3/6	♂ 100%	8–12	RNA	-	-	-
Tsurui [139]	2005	*C57Bl/6*	2/6	♂ 100%	8–12	RNA	-	+	-
Zhao [140]	2017	*BALB/c (H−2d)*	0/6	♂ 100%	6–8	RNA	-	-	-
**90 min ischemia time**									
Ajamieh [141]	2015	*foz/foz and WT*	24	♂ 100%	8	RNA	-	+	-
Duarte [142]	2012	*C57Bl/6 (TIMP −/− and WT)*	6/48/168	♂ 100%	N/A	RNA	-	-	+
Freitas [143]	2010	*C57Bl/6*	6	♂ 100%	8–12	RNA	-	-	-
Hamada [144]	2009	*C57Bl/6 (iNOS −/− and WT) and C57Bl/6 (MMP−9 −/− and WT)*	3/6/24	♂ 100%	8–10	WB	+	-	-
Lee [145]	2016	*C57BL/6*	6	♂ 100%	8	WB	-	-	-
Okaya [146]	2004	*C57Bl/6 (PPARa −/− and WT)*	8	♂ 100%	8–12	WB	+	+	-
Zhou [147]	2020	*C57Bl/6*	6	♂ 100%	8	ELISA/RNA	-	-	-
**Studies not included in meta-analysis**									
**30 min ischemia time**									
Tao [148]	2016	*C57Bl/6*	6	♂ 100%	12	RNA/WB	-	-	-
Lee [122]	2001	*C57Bl/6 (iNOS −/−, eNOS −/− and WT)*	6	♂/♀	N/A	RNA	-	-	-
**40 min ischemia time**									
Patouraux [149]	2014	*C57Bl/6 (Opn −/− and WT)*	4	♂ 100%	10–12	RNA	-	-	-
**75 min ischemia time**									
Kobayashi [150]	2008	*BALB/c*	24	N/A	6–8	WB/IHC	-	-	-
Lee [122]	2001	*C57Bl/6 (iNOS −/−, eNOS −/− and WT)*	6	♂/♀ 50%	N/A	RNA	-	-	-

ELISA, Enzyme-linked immunosorbent assay; IHC, immune histochemistry; N/A, not available; RNA, ribonuclein acid; SP, Spectrophotometric analysis; WB, Western blot; + reported; - not reported.

Figure 4 shows the distribution of warm ischaemia times compared with their reperfusion times for each study of 70% HIRI in mouse models.

Two studies, one with a 100% HIRI model and the other with an unclear ischaemic area, were not included in the meta-analysis (Table 5).

**Table 5 ijms-23-11916-t005:** Other mouse models not included in the meta-analysis.

Model	Reference	Year	Strain	Gender	Age (Weeks)	Reper-fusion Time (h)	iNOS Detection Method	NO Detec.	eNOS Detec.	Survival Rate
100% HIRI 3, 20 min ischemia time	Jessup [151]	2005	*Nude and C57bl/6 (IL-10 −/−, IL-6 −/− and WT)*	♂ 100%	8	4	RNA	+	-	+
70% HIRI + 30% PH	Godwin [152]	2014	*C57bl/6*	♂ 100%	N/A	24	RNA	-	-	-

N/A, not available; PH, Partial hepatectomy; RNA, ribonuclein acid; + reported; - not reported.

Different analysis methods, such as RNA and immunohistochemistry (IHC), were used to measure iNOS We present an overview of the different methods for iNOS detection in Table 6. Additionally, we state the number of valuable results for the meta-analysis for each of these individual methods.

For the meta-analysis, the method of RNA detection was used most frequently, followed by Western blot and IHC. Among all methods, RNA detection could be evaluated for meta-analysis most frequently (13.4%). Enzyme-linked immunosorbent assay (ELISA) and spectrometric detection methods were less likely to be quantified and displayed in the current studies.

### 2.5. Risk of Bias

One hundred and fifteen studies were judged for risk of bias, as shown in Figure 5 and Figure 6. We found that 22.2% of all studies had a low risk by using a random sequence generation, although none of them gave insight into the method used. The risk of bias was equally distributed (40.8% to 43.6%) between low and high risk in baseline characteristics, with a small proportion (16.1%) with unclear risk. Most of the studies (92.0%) gave a statement about animals being housed randomly and equally. For incomplete outcome data, the majority of studies (61.3%) were at high risk, and even 96.4% had no reporting bias. A high risk for other sources of bias was found in 24.3% of the studies because of private funding or analysis errors, with the rest being positively low.

We could not find any statement for allocation concealment, blinding of participants and personal and random outcome assessment in any of the included studies, and these questions were judged as unclear. Two studies did, however, refer to a blinding of outcome assessment with low risk. Apart from histological examinations, none of the other studies referred to any blinded outcome assessment for their data assessment, which meant that the risk of bias was unknown.

### 2.6. Effects of Intervention

For rats and mice, we performed independent meta-analyses for different warm ischaemia models separated into 70% and 100% warm ischaemia on the liver and various warm ischaemia time points. Graphs were grouped by reperfusion time points to show the course.

#### 2.6.1. Analysis of Rat Models with 70% HIRI and 30 min Warm Ischaemia Time

For the studies of rats with 70% HIRI models, three meta-analyses were performed for 30, 60 and 90 min warm ischaemia times.

Four studies reported iNOS values of 70% HIRI and 30 min warm ischaemia time for 60 animals (Figure 7). Considering the reported event rates across studies, we found that three studies had an increased iNOS level when undergoing HIRI. Ischaemia-reperfusion injury caused an increase in iNOS in three studies whereas two studies showed a decrease at a warm ischaemia time of 30 min when compared with the iNOS levels in the sham operation (SHAM) groups for all studies (SMD of 0.51, 95% CI = −3.20–4.21, I^2^ = 93%).

#### 2.6.2. Analysis of Rat Models with 70% HIRI and 60 min Warm Ischaemia Time

Eleven studies reported iNOS values of 70% HIRI and 60 min warm ischaemia time for 159 animals (Figure 8). Considering the reported event rates across studies, we found that 10 studies had an increased iNOS level when undergoing HIRI. Ischaemia-reperfusion injury results caused a large increase in iNOS at a warm ischaemia time of 60 min when compared with the iNOS levels in the SHAM groups for all studies (SMD of 4.81, 95% CI = 3.23–6.397, I^2^ = 81%).

#### 2.6.3. Analysis of Rat Models with 70% HIRI and 90 min Warm Ischaemia Time

Five studies reported iNOS values of 70% HIRI and 90 min warm ischaemia time for 83 animals (Figure 9). Considering the reported event rates across studies, we noted that all studies had an increased iNOS level when undergoing HIRI. Ischaemia-reperfusion injury was caused in a large increase in iNOS at a warm ischaemia time of 90 min when compared with the iNOS levels in the SHAM groups for all studies (SMD of 4.06, 95% CI = 2.29–5.82, I^2^ = 74%).

#### 2.6.4. Analysis of Rat Models with 100% HIRI and 45 min Warm Ischaemia Time

For rats with 100% HIRI, we analysed a 45 min warm ischaemia time in subgroups with different reperfusion time points. The 100% HIRI 30 min and 60 min groups could not be analysed because fewer than three studies showed valid values.

Five studies reported iNOS values of 100% HIRI and 45 min warm ischaemia time for 68 animals (Figure 10). Considering the reported event rates across studies, we identified that four studies showed an increased iNOS level in the animals that were undergoing HIRI. Ischaemia–reperfusion injury caused a large increase in iNOS at a warm ischaemia time of 45 min when compared with the iNOS levels in the SHAM groups for all studies (SMD of 4.53, 95% CI = 1.89–7.16, I^2^ = 82%).

#### 2.6.5. Analysis of Mouse Models with 70% HIRI and 60 min Warm Ischaemia Time

Mice with 70% HIRI and 45, 60 and 90 min warm ischaemia times were analysed in subgroups with different reperfusion time points. A meta-analysis could only be performed in the 60 min group because of an insufficient number of studies (*n* ≥ 3 required by our protocol).

Three studies reported iNOS values of 70% HIRI and 60 min warm ischaemia time for 38 animals (Figure 11). Considering the reported event rates across studies, we noted that all studies showed an increased iNOS level in the animals undergoing HIRI. Ischaemia–reperfusion injury caused a large increase in iNOS at a warm ischaemia time of 45 min when compared with the iNOS levels in the SHAM groups for all studies (SMD of 4.06, 95% CI = 2.58–5.55, I^2^ = 24%). Heterogeneity was moderate because of the small numbers of animals and differences in the detection methods.

Our meta-analysis showed iNOS to be elevated in HIRI versus SHAM-operated animals regardless of species, model or ischaemic time. Our meta-analysis using a random effect showed that iNOS was positively associated with ischaemia damage. The same effects were found for a rat model of 100% HIRI and 45 min ischaemia time. The results of a mouse model of 70% HIRI and 60 min ischaemia time were consistent with the previous results and exhibited low heterogeneity. Overall, the heterogeneity was moderate.

#### 2.6.6. Analysis Comparing NO Values between SHAM and HIRI Groups in Rats

In addition to iNOS, we also investigated NO as a free radical scavenger associated with the iNOS cascade. For the meta-analysis of nitrate/nitrite values, 20 studies were included. Because of missing values, only rat studies of different models were analysed. The studies were grouped by warm ischaemia time for comparison.

Twenty-three studies reported NO values for various HIRI models with different warm ischaemia and reperfusion times for 362 animals (Figure 12). Considering the reported event rates across studies, we estimated that seven studies had decreased NO levels after HIRI. In contrast, 16 studies reported an increase in NO values (SMD of 1.25, 95% CI = −0.00–2.50, I^2^ = 100%). Heterogeneity was high because of the methodological and clinical diversity.

The subgroups within this analysis used different reperfusion time points, which might have affected the results of NO expression. Therefore, we grouped the above results by reperfusion time to determine if there was a trend regardless of the HIRI damage influenced by warm ischaemia time. These meta-analysis results were consistent with the results above.

#### 2.6.7. Analysis of eNOS Values Compared between the SHAM and HIRI Groups in Rats

In addition to iNOS and NO, we performed a meta-analysis to determine if eNOS is affected by HIRI when compared with SHAM-operated animals. Only results for rat studies could be analysed at three different warm ischaemia time points (45, 60 and 90 min).

Eleven studies reported eNOS values for various HIRI models when compared with the SHAM groups with different warm ischaemia and reperfusion times for 156 animals (Figure 13). Considering the reported event rates across studies, there was no significant effect for eNOS for 45, 60 or 90 min warm ischaemia time (SMD −0.78, 95% CI = −2.18–2.90, I^2^ = 90%). The results may have been affected by the low animal numbers, different analysis methodologies (Western blot vs. RNA) and an unknown reperfusion time in the study of Mostafa–Hedeab [41]. Therefore, because of the methodological and clinical diversity, heterogeneity for subgroups was high.

## 3. Discussion

### 3.1. Summary of Main Results

In this present study, we analysed the reporting in 124 animal studies focusing on the relationship between HIRI and iNOS. To the best of our knowledge, this is the first systematic review on the role of iNOS in HIRI. In our quality assessment, studies were divided into subgroups based on outcomes. The literature was subsequently reviewed with a focus on an overview of experimental models, sex distribution, weight or age, warm ischaemia and reperfusion time, iNOS detection method, NOS-related parameters and survival rates for the HIRI and treatment groups with drug treatment.

### 3.2. Rat and Mouse Models within This Systematic Review

This systematic review included rat and mouse models of HIRI with various ranges of ischaemia. Among those HIRI models, an ischaemic area of >60% of liver tissue is clinically relevant. The model of 100% HIRI with portal decompression is ideal for warm ischaemia after PM for hepatic resection and LTx because it best mimics HIRI clinically [153]. The hepatic tolerance to warm HIRI caused by PM is thought to be associated with the duration of liver ischaemia [154]. However, the lack of abundant collateral circulation to the liver in rats and mice limits the ability to assess prolonged ischaemic damage in this 100% HIRI model [153]. The 60–70% HIRI models mimic the conditions for hepatic resection (e.g., posterior segment resection) with occlusion of blood flow in the right lobe. In this systematic review, the 100% HIRI model was found in 23 studies using rat models, and only one study using this technique in a mouse model. Eleven studies could be used in the meta-analysis.

The 70% HIRI model has been widely used in experimental studies of hepatic ischaemia-reperfusion [16]. This technique causes ischaemia by occluding the hepatic artery, portal vein and bile ducts in the left and median lobes. Blood flow through the right and caudal liver lobes allows assessment of prolonged ischaemic damage. In this systematic review, we identified 61 studies that used a 70% HIRI model in rats and 36 studies in mice, of which 40 studies could be used in the meta-analysis.

On the other hand, the 30% HIRI model demonstrated that the blood supply to the right lobe of the liver was interrupted by occlusions at the level of the hepatic artery and portal vein [155]. This model has haemodynamic circulation, which enables the assessment of prolonged ischaemic insults, but it is difficult to apply clinically.

### 3.3. Warm Ischaemia and Reperfusion Models

The standard PM in humans is performed as an ischaemic procedure of the whole liver for 15 min, followed by unclamping for 5 min, and these cycles are repeated until the liver resection is complete [156]. Therefore, a 15 or 20 min ischaemia time at 100% HIRI [28,101,102], 20 min at 70% HIRI [84] or 15 min at 100% HIRI + 70% partial hepatectomy (PH) [114], if PH is added, is in accordance with human clinical practice. However, these studies were not subjected to meta-analysis. The reason for the small number of studies is that the response to ischaemia differs between humans and rodents; thus, the same conditions are not given. So, far, only six studies have used an ischaemia time <20 min. Among all studies (93), the ischaemia time ranged between 30 and 60 min, and 24 studies opted for a higher reperfusion time than 24 h. The ischaemia time is unknown for one study.

Regarding the reperfusion time, the three groups that were included in the meta-analysis had similar sampling points. In the 70% HIRI rat model, iNOS expression was observed from 1 to 24 (1, 2, 4, 5, 6 and 24) h and from 2 to 24 (2, 3 and 24) h after initiation of reperfusion in response to 60 and 90 min of ischaemia, respectively. The 100% HIRI rat model showed iNOS expression from 0.75 to 24 (0.75, 1, 2 and 24) h after initiation of reperfusion in response to 45 min of ischaemia, respectively. In the 100% HIRI mouse model, the expression of INOS was observed from 0.7 to 24 h after the start of reperfusion in response to 45 min of ischaemia. In rat hepatocytes, iNOS mRNA expression at 3 h after inflammatory cytokine stimulation in primary cultured cells, followed by iNOS protein and NO increase at 8 h afterwards, has been reported [157]. Divided into an early phase (0–4 h after HIRI), a middle phase (5–8 h) and a late phase (later), the similarity in the focus on iNOS concentrations can be seen by analysing iNOS mRNA in the early phase, iNOS protein and NO measurements in the middle phase and the associated responses in the late phase. However, the commonality of reperfusion times is impossible to find exactly because of the different conditions of the models and the different focus of the authors.

In this systematic review, we sought to determine if a certain ischaemia time is associated with a reperfusion time so that we could determine if a mouse or rat model is best suited to investigate the iNOS pathways. Unfortunately, we showed that the warm ischaemia time and reperfusion time varied between mouse and rat models of the same HIRI. A standardisation for comparing the outcomes is needed not only with regard to animal models but also to clinical situations and possible translation of results.

### 3.4. Overall Completeness and Applicability of Evidence

#### 3.4.1. Sex Distribution

Regarding the general characteristics, various experiments have been conducted without standardisation. We also observed a sex bias in our analysis. In our selected studies, mostly male mice and rats (97.4%) were used [158]. Female rats have a 4-day sexual cycle, which would affect the experiment because oestrogen has a hepatoprotective effect [159]. Sex dimorphism in hepatic ischaemia-reperfusion in the murine liver ischaemia-reperfusion model has been demonstrated [160]. It was found that 100% of the female mice survived indefinitely whereas all male mice died within 5 days. The protective effect in females appeared to be due to ovarian oestrogen, as ovariectomy of females or administration of a selective oestrogen antagonist to females resulted in an enhanced liver injury and greater mortality rate. It was shown that oestrogen protects against ischaemic-reperfusion injury of the liver, which was associated with increased levels of serum NO and decreased levels of TNFα [161]. In another animal study, the mechanism of the effects of oestrogen included its antioxidative nature [162,163,164]. These findings might explain why the studies in this systematic review opted to use males. In humans, different mechanisms have been hypothesised, including the effect of sex hormones on oxidative, metabolic and immunological pathways, as well as differential gene transcription in response to acute and chronic liver injury [165]. According to the European Liver Transplant Registry data, 46,334 patients underwent liver transplant in Europe between December 2002 and December 2012, with 32,656 (70.5%) males and 13,678 (29.5%) females [166]. In 2011–2012, the proportion of male donors was 56.1% and female donors was 43.9% [166]. Thus, ischaemia-reperfusion injury can occur in both males and females, but most studies in rodents have been exclusively conducted in males. Additionally, there are sex differences in the liver metabolism of human cytochrome P450 isoforms in rodents [167,168]. It is important to study novel therapies at the basic science level in both sexes [169]. Therefore, it is quite possible that studies using only males may not translate into certain clinical situations.

#### 3.4.2. iNOS Generation and Function

Transcriptional regulation is the main control of iNOS expression. In general, proinflammatory cytokines, such as TNF-alpha, IL-1, interferon-gamma and lipopolysaccharides, first bind to cell surface receptors and activate kinases, leading to the phosphorylation of various intracellular proteins. Specific transcription factors, including nuclear factor κB and transcription factors, such as nuclear factor 1 transducer and activator of transcription 1a, are activated. The activated factor translocates into the nucleus, binds to the promoter region of the iNOS gene and induces expression of iNOS [170].

Overexpression or deregulation of iNOS causes excessive NO levels, which can lead to toxic effects and are associated with a number of human diseases, such as septic shock, cardiac dysfunction, pain, diabetes and cancer [171]. Large amounts of NO play an important role in the inflammatory response and the innate immune system by helping to protect against incoming pathogens [172]. We speculate that decreased iNOS in humans would reduce inflammatory responses and defences against the innate immune system, but this has not been reported.

In this systematic review, iNOS was found to have been measured by a variety of analytical methods in mRNA and protein (Western blot, IHC, ELISA and spectrometry). The heterogeneity of iNOS assays made comparisons difficult. However, mRNA assays are widely used (13.4%) and provide quantitative comparability, but they cannot prove that iNOS is transcribed instead of metabolised. Therefore, we require a more uniform measurement method with standardised scaling focusing on the presence of the effective protein.

#### 3.4.3. iNOS in HIRI versus SHAM

High NO concentrations due to the promotion of iNOS may lead to harmful products, such as peroxynitrite (ONOO−), and other reactive oxygen species (ROS) are formed [173]. ROSs cause not only direct damage to cells, but also activate a cascade of molecular mediators, leading to acute inflammatory changes with microvascular alterations, increased apoptosis and increased necrosis of liver cells [174]. In this study, we found that HIRI increased iNOS in rats and mice without transgenic backgrounds. In iNOS knockout mice, mRNA of iNOS was not expressed at reperfusion [121].

#### 3.4.4. Increased iNOS May Induce Hepatic Damage Via Significant Production of ROS

Within our meta-analysis, we demonstrated that iNOS was elevated when HIRI was compared with SHAM regardless of species and warm ischaemia time. This is consistent with the common knowledge that iNOS is upregulated after surgery with such high impact on the liver.

### 3.5. Agreements and Disagreements with Other Studies or Reviews

#### 3.5.1. Nitric Oxide in HIRI Models versus SHAM

In human clinical practice, it has been reported that NO was produced in the liver during HIRI and that iNOS is involved in the production of NO. Plasma nitrate and nitrite were determined by the Griess reaction [175]. In addition to iNOS, a meta-analysis was performed to determine if NO is affected by HIRI relative to animals undergoing SHAM surgery. The present analysis did not show any effect on NO in HIRI, or any relationship with iNOS, which may be because of the lack of standardisation of conditions (e.g., reperfusion time and ischaemia time) and NO detection (method and time) in animal experiments.

#### 3.5.2. eNOS in HIRI Models versus SHAM

The hepatoprotective effect of eNOS against HIRI is related to increasing NO levels, and inhibition of eNOS overexpression also protects against HIRI. Multiple eNOS/NO signalling pathways (e.g., Akt-eNOS/NO, AMPK-eNOS/NO, HIF-1α-eNOS/NO) participate in the anti-HIRI process [31].

The effect of NO on HIRI has been reported as potentially harmful, beneficial or both [176]. The NO from eNOS is considered to have protective effects and to maintain homeostasis [177,178]. On the other hand, high NO production from iNOS has potentially toxic effects because it may react with superoxide anions to form toxic peroxynitrite [179,180]. In a study of HIRI using genetically engineered mice, iNOS knockout mice showed decrease [122], increase [119] or no change [121] in toxicity caused by NO. In addition, eNOS mRNA was strongly expressed at 1 h post-perfusion in iNOS-knockout mice [120]. From these reports, the role of iNOS and eNOS in HIRI, as well as their interaction with each other, remains unclear. This systematic review excluded genetically modified mice. Therefore, we were not able to further investigate the role of eNOS in HIRI.

The meta-analysis of eNOS of animals undergoing HIRI versus SHAM operation did not show any effect, and thus did not reveal any relationship with iNOS whatsoever.

#### 3.5.3. iNOS and Survival

The indications for reducing iNOS are ischaemia-reperfusion injury, sepsis and pain [181]. Attempts to treat ischaemia-reperfusion injury and septic shock with drugs intended to reduce iNOS in clinical practice have been largely unsuccessful and none have been used in clinical practice [182]. One animal study found that the iNOS inhibitor 2-aminoethyl-isothiourea [78] may not have benefits for HIRI universally. ONO-1714 has been used to reduce HIRI [75]. The non-specific NOS inhibitors, NG-nitro-L-arginine (L-NNA) [183] and L-NAME [34], have been shown to increase HIRI. On the other hand, L-NAME has shown a hepatoprotective effect by inhibiting iNOS mRNA on HIRI after portal triad clamping and reperfusion for nonselective inflow occlusion and partial hepatectomy of cholestatic livers induced by bile duct ligation in rats [110]. L-NAME may be protective or harmful, depending on the model and timing of the experiment, which might be explained by L-NAME being a nonselective NOS inhibitor and by competitive inhibition. L-NAME is a potent inhibitor not only of iNOS but also of eNOS. The use of nonselective NOS inhibitors in clinical practice may increase side effects because eNOS is also inhibited. On the other hand, selective inhibition of iNOS is expected to have fewer side effects, but it is currently still under investigation.

Surgical approaches to ischaemia-reperfusion injury, aimed at reducing iNOS, also have not been used in clinical practice. Evidence of the effect of intermittent hepatic ischaemia (IPC) on NO from eNOS and iNOS is limited. IPC is considered to protect short-term ischaemia against subsequent long-term ischaemic injury. Koti et al. and Guo et al. reported that eNOS was upregulated in rats receiving liver IPC + HIRI compared to rats receiving HIRI [21,125]. The increased eNOS expression was associated with significantly higher plasma NOx levels in the IPC + HIRI group than in the HIRI group. Expression of iNOS was not found in all experimental groups, including both liver IPC + HIRI and HIRI groups [21,125]. On the other hand, Longo et al. reported that the expression of eNOS and iNOS was not increased in the livers of rats receiving intrahepatic IPC + HIRI relative to the levels in rats receiving HIRI [74]. In remote IPC, increased NO due to eNOS expression in the liver [81] and iNOS in skeletal muscle is thought to act against HIRI protection.

A study of increased iNOS mRNA by administration of metron factor-1 (MF-1) reported increased survival. In HIRI, MF-1 improved hepatic microcirculation by increasing NO expression, and NO-induced damage was not obvious [105]. In these studies, focusing on survival experiments, there was no evidence of a link between increased survival and changes in iNOS alone. Apart from NOS-related and general (e.g., liver function enzymes, inflammatory cytokines, pathology) analyses, there were no analyses in the literature focused on other mechanisms unaffected by iNOS. In discussion, the overall improvement in the condition was considered to have contributed to the increased survival rate.

Another strategy to improve the survival rate is to avoid HIRI in the first place. New surgical tools are available to dissect liver tissue without the need for a Pringle manoeuver, thus preventing ischemia-reperfusion injury. Such instruments as the Habib sealer or ultrasonic scalpels can coagulate the tissue while cutting [184,185]. Studies in patients showed fewer postoperative complications and a reduction of blood loss as well as reduced liver manipulation and significant reduced warm ischemia time [186,187,188].

Considering the fact that many included studies investigated reducing iNOS expression in HIRI models, it is interesting that there is a lack of studies linking the iNOS reduction to a possible prolonged survival.

### 3.6. Potential Biases in the Review Process

Our systematic review had several limitations. The aim of this systematic review was to provide a wide overview into current approaches and the state of research on the effects of warm ischaemia and reperfusion injury in mouse- and rat-liver models on iNOS and other related parameters. Our systematic search was therefore designed to be broad and include as many related studies as possible. Consequently, we retrieved a variety of differing approaches and were able to qualitatively assess them. Nonetheless, this very inclusive approach might have also precluded some results that might have been obtained if we had used a search protocol that more specifically focused on some of the subsets we identified. Although we are confident in presenting a somewhat comprehensive summary of the review question, it is possible that due to the broadness of our search protocol some eligible studies were not covered. Although we have searched three independent databases it is possible that the search in Web of Science is limited due to licensed access. Additionally, this database has no equivalent to MeSH-terms and we might have missed some studies. One limitation was our exclusion of non-English literature. Another limitation is the fact that we performed the meta-analysis exclusively for HIRI versus SHAM-operated animals. We were therefore not able to implement all included studies as some did not use a SHAM model in their experiments.

Unfortunately, we were not able to include many studies due to incomplete data, such as animal number. We tried to reach out to the authors at least twice but received very little response. We are also aware of the possibility that negative results have not been published. As we included only studies with available data for iNOS, this might have limited the outcome. Additionally, we were only able to implement results if a table or graph provided them. In the case of immunohistochemistry, we were barely able to carry out counting because only example pictures have been reported. Additionally, many publishers limit their word count, so the authors may have had to adjust the length of their reports and exclude some relevant elements. Another limitation is that we could not adjust for the reporting quality of the studies as it related to word count limits of the publishing journals because of some practical obstacles (changes in publication policies over the years, difficulties in contacting every single publisher, and the fact that several journals have been sold or the publisher has changed over the decades); however, we have included the available online supplements in our assessment. Nonetheless, interpretation of these findings must be performed carefully. We cannot rule out the possibility that certain confounders that were not accounted for in the analysis contributed to the observed significant result. Due to the high risk of bias in incomplete outcome data, as well as an unclear randomised assessment of data, we cannot rule out a compromised scientific interpretation of study results.

## 4. Materials and Methods

This systematic review and meta-analysis were conducted according to the Cochrane standard and was reported according to the principles of the Preferred Reported Items for Systematic Reviews and Meta-Analyses (PRISMA) guideline. The PRISMA checklist can be found in the Appendix A. The protocol is available at PROSPERO (registration number CRD 42020191298).

### 4.1. Search Strategy

A systematic literature search was performed encompassing PubMed (1946–April 2022); [title/abstract], [MeSH]), Web of Science (1990–April 2022) and Embase databases (1947–April 2022; [ti.ab], [exp/mj]). This literature search was performed to identify all potentially relevant articles on triggered liver regeneration after warm ischaemia in mouse and rat models with a focus on iNOS. The applied search terms are shown in Figure 14. The search terms were limited to Title/Abstract in PubMed and adjusted accordingly for the other two databases. These results were imported to Endnote Version X8 (Clarivate Analytics, NY, USA). Endnote was used to identify duplicates and to generate an Excel file for independent screening and comparison thereafter. Since Web of Science has licensed access, the available collections for this systematic review were:Web of Science Core Collection Indexes;Science Citation Index Expanded (SCI-Expanded) (1900-);Social Sciences Citation Index (SSCI) (1900-);Arts & Humanities Citation Index (A&HCI) (1975-);Conference Proceedings Citation Index—Science (CPCI-S) (1990-);Conference Proceedings Citation Index—Social Sciences & Humanities (CPCI-SSH) (1990-);Book Citation Index—Science (BKCI-S) (2005-);Book Citation Index—Social Sciences & Humanities (BKCI-SSH) (2005-);Current Chemical Reactions (CCR-Expanded) (1985-);Index Chemicus (IC) (1993-);BIOSIS Previews (1926-): Journals, patents and conference proceedings in biomedicine;MEDLINE (1950-);Russian Science Citation Index (2005-);SciELO Citation Index (1997-);Journal Citation Reports.

### 4.2. Searching Other Resources

We identified other potentially eligible studies by searching the reference lists of systematic reviews and the 124 included studies.

### 4.3. Criteria for Considering Studies for This Review

The study selection was based on the following inclusion and exclusion criteria.

#### 4.3.1. Types of Studies

We included research publications in international peer-reviewed literature (including inter-library loan requests, open access and closed access) available as full-text publications in English. Conference abstracts, review articles, case studies, editorials, withdrawn articles and those in languages other than English were excluded.

#### 4.3.2. Types of Species

Experimental animal studies using living mice or rats were included in this systematic review. All other species and iNOS or eNOS knockout rats and mice were excluded. A search for other species revealed that mice and rats were the only rodents used in combination with our search terms. A further search listed only twelve studies in which pigs and dogs were used as large animal models combined with iNOS and HIRI.

#### 4.3.3. Types of Intervention

Experimental animal studies that performed laparotomy and warm HIRI with and without partial hepatectomy under anaesthesia were included if warm ischaemia was applied to the whole or parts of the liver graft. Studies were also included if they performed partial hepatectomy and if techniques, such as PM, were used during which the graft underwent warm ischaemia. Liver transplantation was excluded because of the effect on the organ and body yielding various immune system responses. Additionally, grafts for transplantation that received cold ischaemia during storage combined with warm ischaemia afterwards were excluded. Therefore, the liver transplantation data will not be comparable to data from animals that underwent simple warm ischaemia following reperfusion. Additionally, partial hepatectomy without warm ischaemia and reperfusion and partial hepatectomy >70% resection were excluded, as were ex vivo studies and those involving treatment for warm ischaemia and reperfusion with drugs or ischaemic preconditioning.

#### 4.3.4. Types of Comparison

We included a comparison of animals that underwent sham operations versus animals that underwent warm ischaemia and reperfusion, as well as control animals versus animals with various treatments for meta-analysis. Studies without SHAM or HIRI controls were excluded.

Types of outcome measures included were all studies with the parameter of iNOS measured in liver, serum or plasma. If no detection of iNOS was made the study had to be excluded.

### 4.4. Outcomes

The following outcomes were assessed by meta-analysis:We compared iNOS values of SHAM to HIRI groups using subgroups of different reperfusion times:Rat 70% HIRI at 30 min warm ischaemia time, data grouped by reperfusion time;Rat 70% HIRI at 60 min warm ischaemia time, data grouped by reperfusion time;Rat 100% HIRI at 45 min warm ischaemia time, data grouped by reperfusion time;Mouse 70% HIRI at 60 min warm ischaemia time, data grouped by reperfusion time.Effects of parameters related to iNOS:NO parameters of SHAM groups versus HIRI groups in rat 70% HIRI data grouped by warm ischaemia times;eNOS parameters of SHAM groups versus HIRI groups.

### 4.5. Differences between Protocol and Review

We specified data handling within the meta-analysis by adding subgroups to distinguish between different reperfusion times for Figure 7, Figure 8, Figure 9, Figure 10, Figure 11, Figure 12 and Figure 13. Furthermore, we did not investigate into survival rates. The protocol states that a formal screening was performed while the preliminary searches were yet not finished. We want to clarify that this was a peer-review screening and the results of this screening were implemented into the systematic review if the studies met our final inclusion criteria.

### 4.6. Data Collection and Analysis

Two authors (RN, MS) independently screened the results of the search strategies for eligibility for this review by reading the titles and abstracts using EndNote Software (EndNote X8, Clarivate Analytics, London, UK). We coded the abstracts as either ‘include‘ or ‘exclude‘. In the case of disagreement, or if it was unclear whether we should include the abstract or not, we obtained the full-text publication for further discussion. Both review authors assessed the full-text articles of the selected studies. Disagreements concerning the suitability of an article were discussed by the study team. A third reviewer was not needed as all disagreements were resolved. If applicable, online Appendix A was also retrieved for further analysis.

We documented the study-selection process in a flow chart, as recommended in the PRISMA statement [189], and show the total numbers of retrieved references and the numbers of included and excluded studies.

### 4.7. Handling of Missing Data

We contacted authors for missing data at least twice to confirm study characteristics and obtain details for the risk of bias assessment. If studies only showed their results in graphs, we used a digital ruler to measure the data from the graph to the best of our ability if possible.

The following data were extracted from each of the included studies:Species/strain and liver model;Warm ischaemia and reperfusion time for the HIRI and SHAM groups;Sex distribution of mice and rats for each study;Weight or age of experimental animals for each study;iNOS detection method;NO parameter and eNOS parameters assessed parallel to iNOS for the SHAM and HIRI groups.

### 4.8. Risk of Bias and Quality Assessment

Two authors (R.N., M.S.) independently evaluated the methodological quality of the included studies using the risk of bias tool for animals according to the Systematic Review Center for Laboratory Animal Experimentation (SYRCLEs) risk of bias. Two different authors (C.K., S.B.) performed the risk of bias assessment for the studies by Ishizaki [84] and Iwasaki [110] to avoid any conflicts of interest.

We adapted the tool based on the needs of this systematic review and meta-analysis as recommended by SYRCLEs [190]. The following aspects were covered in our study’s risk of bias tool: (a) random sequence generation, (b) baseline characteristics, (c) allocation concealment, (d) random housing, (e) blinding of caregivers and investigators, (f) random outcome assessment, (g) blinding of outcome assessors, (h) incomplete outcome data, (i) selective reporting and (j) other bias.

### 4.9. Data Analysis and Statistics

Data were compared and plotted using Review Manager Software (Version 5.3, Nordic Cochrane Center, Copenhagen, Denmark). We performed the meta-analysis only when enough data (*n* ≥ 3) were available. In tests for overall effect, *p* < 0.05 was accepted as indicative of statistical significance. A random-effects model for the meta-analysis was chosen to compensate for methodological experimental heterogeneity such as parameter assessment, as well as various treatments. For group size, the mean values were used. Continuous data are presented as the standardised mean difference (SMD) with 95% confidence intervals (CI) as studies used different scales to measure the same outcome. Heterogeneity was tested using I^2^ [191].

## 5. Conclusions

Our systematic review and meta-analysis showed that iNOS was clearly increased in HIRI in both mice and rats regardless of the warm ischaemia time or reperfusion time (end point). We did not identify any association between iNOS upregulation and an effect on NO or eNOS levels.

We found a lack of comparability of different detection methods for iNOS leading to a decreased number of studies including into meta-analysis. Additionally, the number of animals per group was often unclear. We recommend that studies need to be more precise and publish complete results for later evaluation.

We were not able to find a direct link between iNOS reduction and prolonged survival within the literature. However, further investigation into the causality of a possible association of iNOS with survival is needed. Furthermore, the specific circumstances that determine if iNOS-generated NO will have a positive effect on survival also require more study.

## Figures and Tables

**Figure 1 ijms-23-11916-f001:**
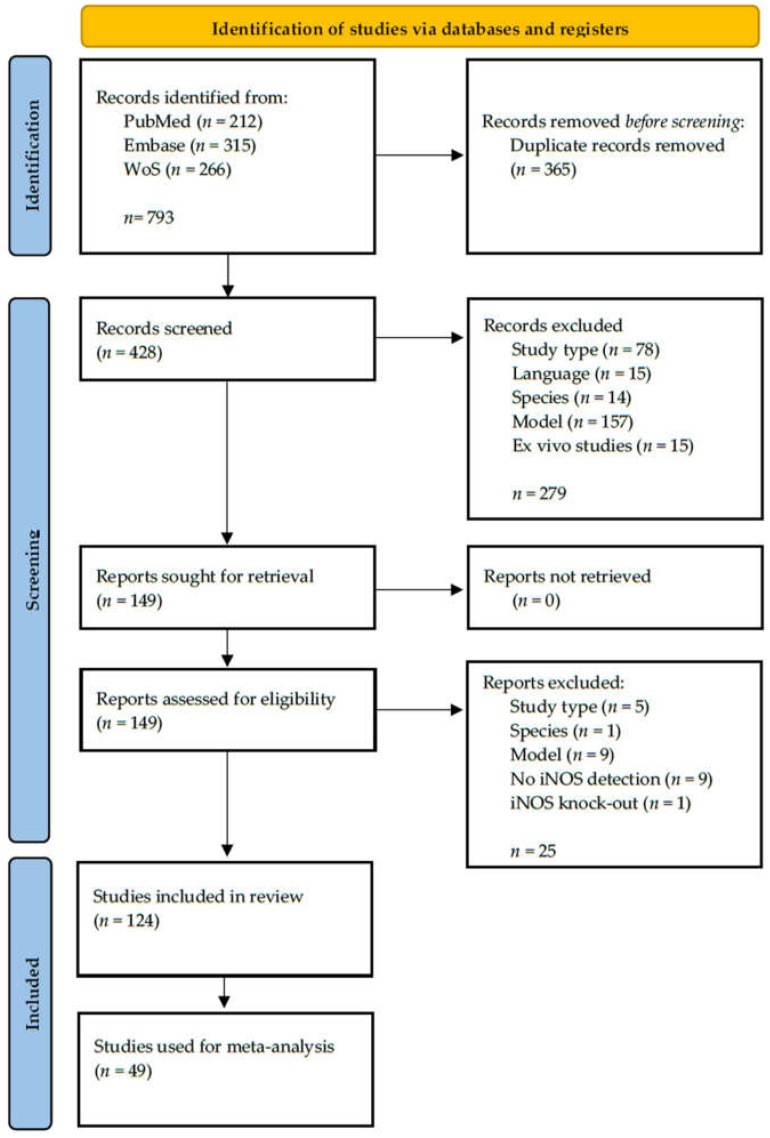
Flow chart of literature selection.

**Figure 2 ijms-23-11916-f002:**
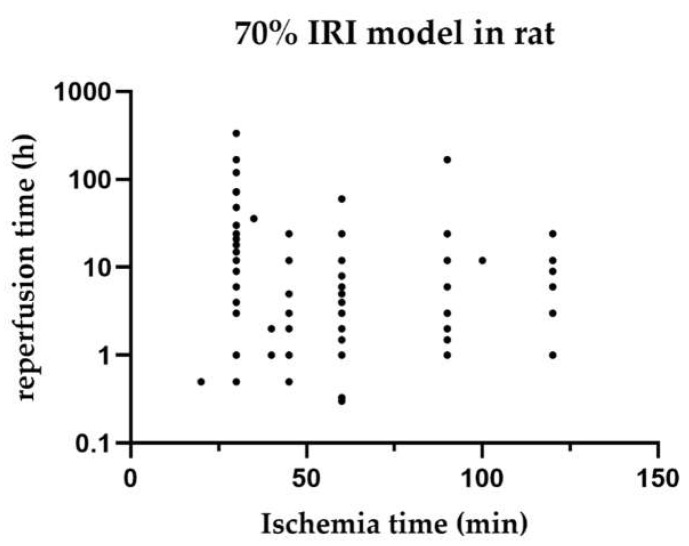
Distribution of reperfusion times compared with their ischaemia times in 70% ischaemia models in rats. Studies with various reperfusion times are listed for each individual time point. Time points are listed regardless of the survival time of animals.

**Figure 3 ijms-23-11916-f003:**
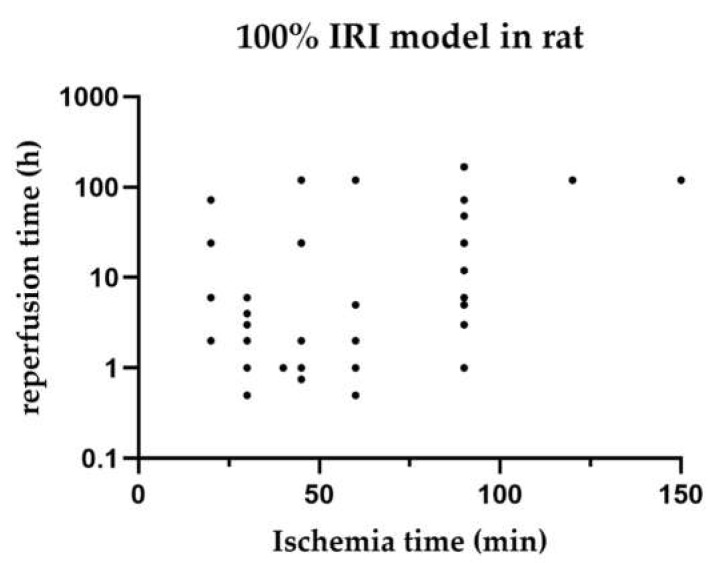
Distribution of reperfusion time compared with their ischaemia times in 100% ischaemia models in rats. Studies with various reperfusion times are listed for each individual time point. Time points are listed regardless of the survival time of animals.

**Figure 4 ijms-23-11916-f004:**
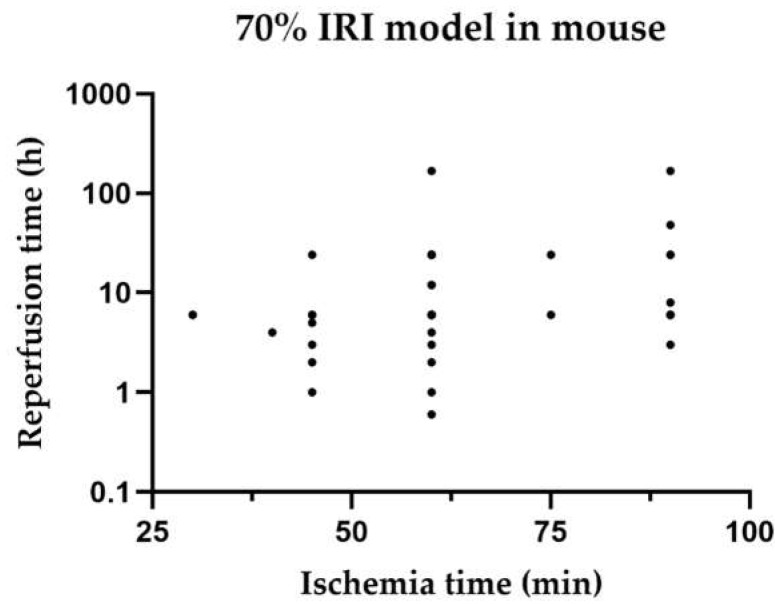
Distribution of reperfusion times compared with their ischaemia times in 70% ischaemia models in mice. Studies with various reperfusion times are listed for each individual time point. Time points are listed regardless of the survival time of animals.

**Figure 5 ijms-23-11916-f005:**
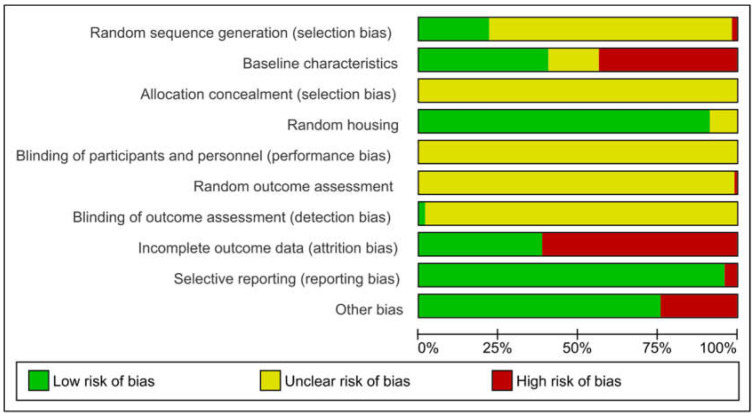
Risk of bias assessment for the 125 studies included.

**Figure 6 ijms-23-11916-f006:**
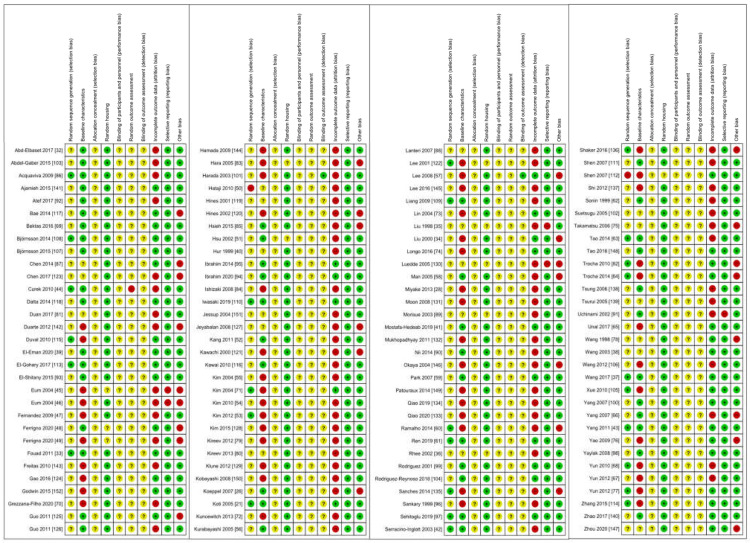
Detailed risk of bias assessment for all 10 signalling questions.

**Figure 7 ijms-23-11916-f007:**
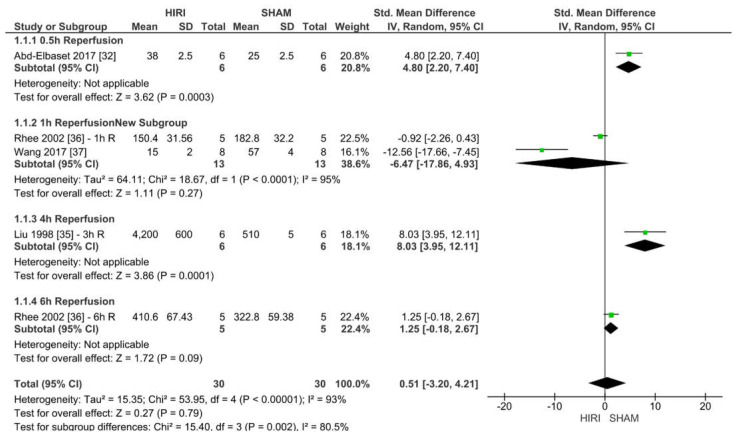
Forest plot of 70% HIRI in rats and 30 min warm ischaemia time grouped by reperfusion time between 0.5 and 72 h.

**Figure 8 ijms-23-11916-f008:**
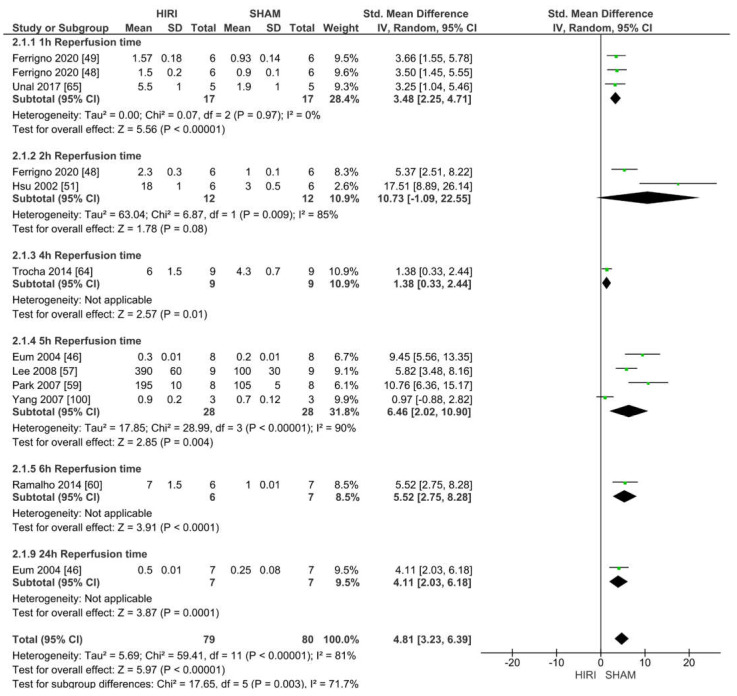
Forest plot of 70% HIRI in rats and 60 min warm ischaemia time grouped by reperfusion time between 1 and 24 h.

**Figure 9 ijms-23-11916-f009:**
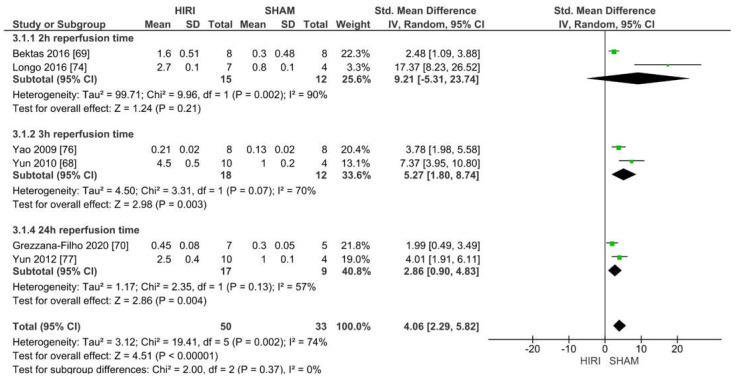
Forest plot of 70% HIRI in rats and 90 min warm ischaemia time grouped by reperfusion time between 2 and 24 h.

**Figure 10 ijms-23-11916-f010:**
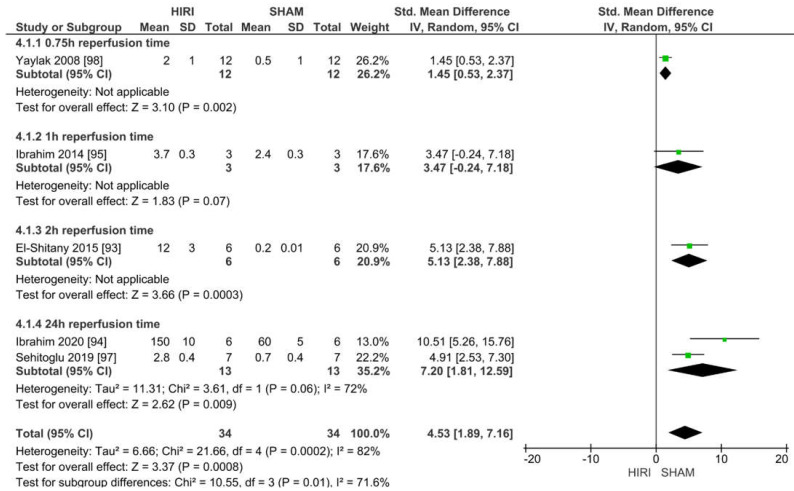
Forest plot of 100% HIRI in rats and 45 min warm ischaemia time grouped by reperfusion time between 0.75 and 24 h.

**Figure 11 ijms-23-11916-f011:**
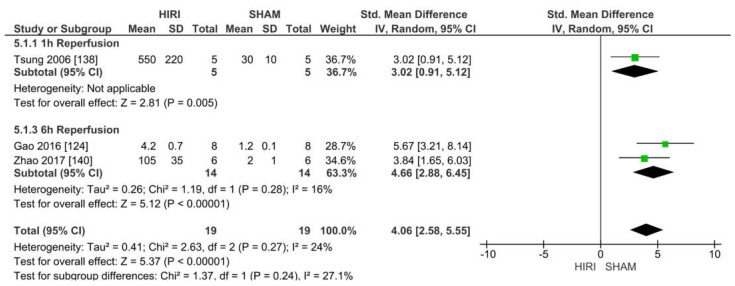
Forest plot of 70% HIRI in mice and 60 min warm ischaemia time grouped by reperfusion time between 1 and 6 h.

**Figure 12 ijms-23-11916-f012:**
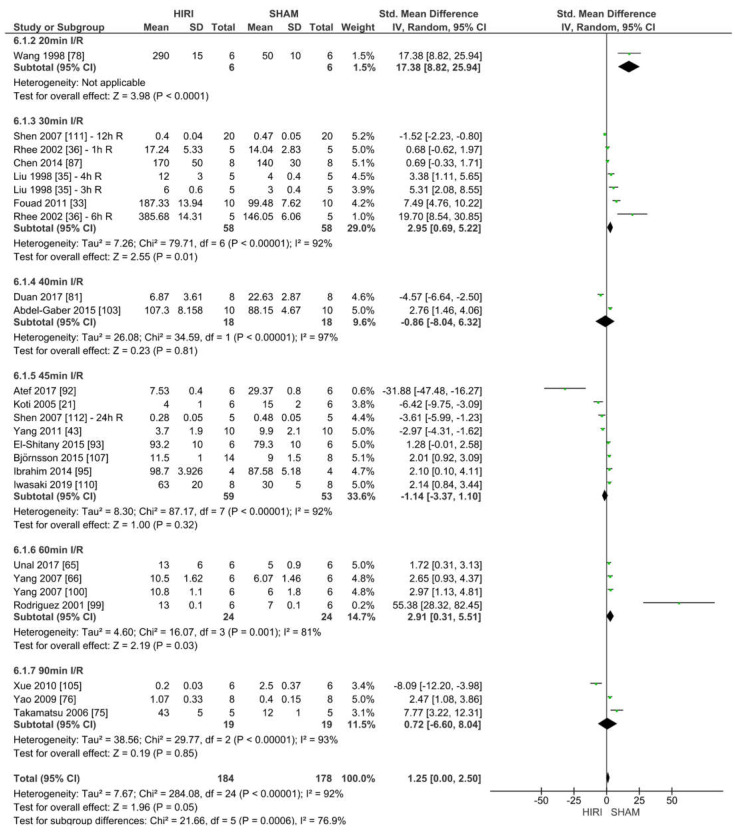
Forest plot of NO detection in SHAM groups versus HIRI grouped by different warm ischaemia durations.

**Figure 13 ijms-23-11916-f013:**
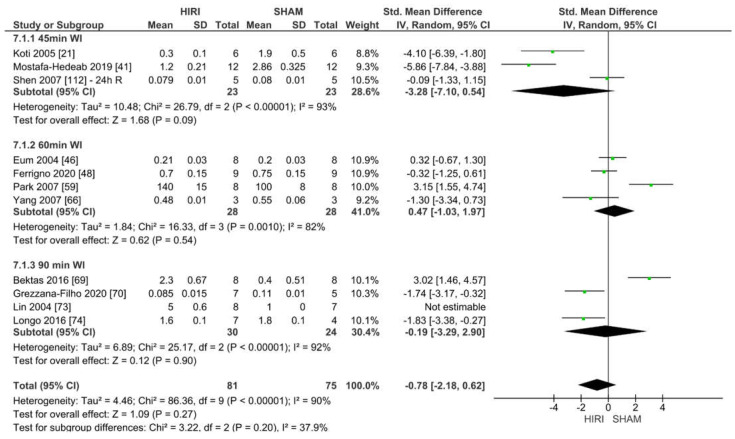
Forest plot of eNOS detection in SHAM versus HIRI grouped by different warm ischaemia durations.

**Figure 14 ijms-23-11916-f014:**
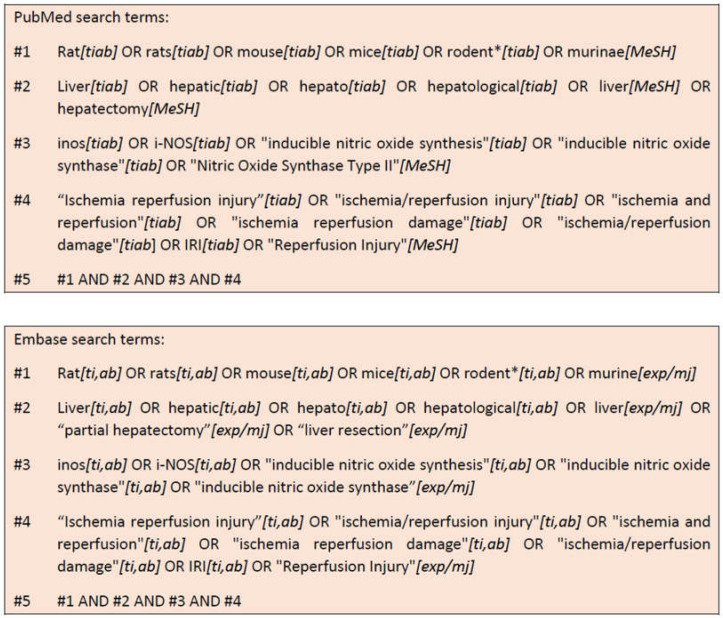
Search terms and their combination used for PubMed and Embase.

**Table 6 ijms-23-11916-t006:** Methods of iNOS detection and number of methods valuable for meta-analysis.

Method	N° of Used Method	N° Used for Meta-Analysis	% of Used Methods
ELISA	7	3	2.2
IHC	24	8	6.0
RNA	61	18	13.4
Spectrometry	5	3	2.2
WB	37	8	6.0
Total	134	40	29.8

ELISA, Enzyme-linked immunosorbent assay; IHC, immune histochemistry; RNA, ribonuclein acid; SP, Spectrophotometric analysis; WB, Western blot.

## Data Availability

The data of this study are openly available in ‘Zenodo’ at https://doi.org/10.5281/zenodo.5082151 (accessed on 8 July 2021).

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
