# Peer review of "Effects of iNOS in Hepatic Warm Ischaemia and Reperfusion Models in Mice and Rats: A Systematic Review and Meta-Analysis"

_ijms, 2022, doi:10.3390/ijms231911916_

Round 1

Reviewer 1 Report

Nakatake et al present a meta-analysis of rodent studies featuring liver warm ischemia, examining the effect of ischemia and reperfusion times on the expression of iNOS, eNOS and the production of NO.

While a tremendous amount of work went into this paper, there are major points to address before considering it for publication.

1.     The title is too encompassing. While extensive, this review focuses on iNOS, eNOS and the production of NO. This is not enough to qualify at the “Effects of Warm Ischaemia and Reperfusion in Mouse and Rat Liver Models: A Systematic Review and Meta-Analysis”

2.     Regarding the objectives, the authors clearly answer to the first one “the knowledge gained from most single studies is limited”, but they do not provide answers for “It remains unknown if NO induced by iNOS may has a protective or a deleterious effect during warm HIRI.” And “the relationships between iNOS and other proinflammatory cytokines and factors in HIRI have not been clarified.”

Hence the objectives may need to be rectified

3.     The conclusion that “Our systematic review and meta-analysis showed that iNOS was clearly increased in HIRI in both mice and rats regardless of the warm ischaemia time or reperfusion time (end point).” Is true but lack relevance as this is a well-known fact. The authors may want to discuss the impact of the “regardless of the warm ischaemia time or reperfusion time” in clinical practice: the concept that iNOS, or NO, could be used to evaluate the impact of a particular ischemia (through the pringle maneuver) on the liver is thus not a sound one, which should lead towards either looking for other markers of severity, or strategies to limit ischemia altogether. The latter option seem to be the one chosen in surgery (at least in France) as technical innovations are being tested towards decreasing the use of the pringle maneuver and exploring other ways to limit hemorrages. A paragraph on this point would increase relevance of this work.

Minor:

-       All iNOS figs should be like fig 12 & 13, not need for repeated figures

Reviewer 2 Report

General comments

The MS No.: [IJMS] Manuscript ID_ijms-1935786: Effects of Warm Ischaemia and Reperfusion in Mouse and Rat Liver Models: A Systematic Review and Meta-Analysis by authors: Richi Nakatake, Mareike Schulz, Christina Kalvelage, Carina Benstoem and René H. Tolba, represent a review article and meta-analysis based on overview of three database including 124 studies from which 49 records were used in the meta-analysis, following inducible nitric oxide synthase (iNOS) as an early inflammatory response to warm ischaemia induced by Pringle Manoeuver (which is used to minimize bleeding and simultaneously induce warm ischemia) during hepatectomy in hepatic ischaemia reperfusion injury (HIRI)  in mouse and rat liver models. Authors excluded from the study rat HIRI models with procedures such as ischaemic preconditioning, cirrhotic liver, atrial hepatectomy or splenic–caval shunt, because of significant methodological heterogeneity with regard to the experimental model. Author also included thirty-nine studies in mouse models. 

The title of MS is clear and adequate.

Abstract is well written, clear and and self-explanatory for the readers.

The introduction section is well written and explains the basic concepts of HIRI and the influence of iNOS as an early inflammatory response to HIRI.

The manuscript concept is well designed and based on the fact that NO and its derivatives are known to have important roles in the pathophysiology of the liver. It is known that, in contrast to eNOS, which has a protective influence, iNOS contributes to pathological processes by acting as a proinflammatory mediator. Thus, the MS provides enough basic information about the mechanisms of action of iNOS in conditions of induced HIRI.

Materials and methods are based on the well performed meta analysis of carefully selected cases of rat and mouse HIRI.

Analysis is  well and clear presented with very clear and representative illustrations and detailed explanations of including/excluding criteria.

Authors noted that all studies showed a large increase in iNOS level in the animals undergoing HIRI at warm ischemia time of 45 min. Meta-analysis also showed elevated iNOS in HIRI versus SHAM-operated animals regardless to species, model or ischaemic time. Authors also presents analysis comparing NO values between SHAM and HIRI groups in rats.

In summary, obtained results suggest the importance of HIRI prevalence which is of grate interest for regeneration after liver operation. Authors stated that this study is the first systematic review on the role of iNOS in HIRI. It is important to note that mostly male mice and rats (97.4%) were used. Female rats have a 4-day sexual cycle, which would affect the experiment because estrogen has a hepatoprotective effect. It was found that 100% of the female mice survived indefinitely whereas all male mice died within 5 days. In humans, different mechanisms have been hypothesized, including the effect of sex hormones on oxidative, metabolic and immunological pathways, as well as differential gene transcription in response to acute and chronic liver injury. In humans, ischaemia-reperfusion injury can occur in both males and females, but most studies in rodents have been exclusively conducted in males. Also, there are sex differences in the liver metabolism of human cytochrome P450 isoforms in respect to rodents.

Authors also stated that their study had several limitations, for example that using current protocol some eligible studies were not covered. Under the given circumstances, authors cannot rule out a compromised scientific interpretation of study results. As the authors stated, it is of importance to further explore the underlying molecular mechanism in more depth.  

Finally, the review represents a contribution to the overall scientific knowledge in this area and provides a solid basis for further analyses.

I have no specific comment in the text.

Having all the above in mind, I suggest to the editor to accept this review manuscript for publication in the present form.

My final opinion: acceptable for publication.

Author Response

We sincerely thank the reviewer for his comprehensive evaluation of our manuscript. We appreciate his positive assessment. We have made slight changes according to Reviewer 1, including adjusting the title and objective to make them more precise. Additionally, we have added one small paragraph within the Discussion to highlight recent studies on shortening the warm ischemia time during surgery.
We hope that the reviewer agrees with these changes.